# Prevalence and determinants of precancerous cervical lesions among women screened for cervical cancer in Africa: A systematic review and meta-analysis

Berihun Agegn Mengistie[1]*, Getie Mihret Aragaw[1], Tazeb Alemu Anteneh[2], Kindu Yinges Wondie[2], Alemneh Tadesse Kassie[2], Alemken Eyayu Abuhay[3], Wondimnew Mersha Biset[4], Gebrye Gizaw Mulatu[5], Nuhamin Tesfa Tsega[6]

1 Department of General Midwifery, School of Midwifery, College of Medicine and Health Sciences, University of Gondar, Gondar, Ethiopia, 2 Department of Clinical Midwifery, School of Midwifery, College of Medicine and Health Sciences, University of Gondar, Gondar, Ethiopia, 3 University of Gondar Comprehensive Specialized Hospital, Gondar, Ethiopia, 4 Department of Anesthesiology, Critical Care and Pain Medicine, Saint Paul's Hospital Millennium Medical College, Addis Ababa, Ethiopia, 5 Department of Health Informatics, Institute of Public Health, College of Medicine and Health Sciences, University of Gondar, Gondar, Ethiopia, 6 Department of Women's and Family Health, School of Midwifery, College of Medicine and Health Sciences, University of Gondar, Gondar, Ethiopia

* berihunagegn21@gmail.com

## Abstract

### Background

Precancerous cervical lesions, or cervical intraepithelial neoplasia (CIN), represent a significant precursor to cervical cancer, posing a considerable threat to women's health globally, particularly in developing countries. In Africa, the burden of pre-malignant cervical lesions is not well studied. Therefore, the main purpose of this systematic review and meta-analysis was to determine the overall prevalence of pre-cancerous cervical lesions and identifying determinants among women who underwent cervical cancer screening in Africa.

### Methods

This study followed the Preferred Reporting Item Review and Meta-analysis (PRISMA) guidelines. The protocol for this systematic review and meta-analysis was registered on the International Prospective Register of Systematic Reviews (PROSPERO) (ID: CRD42025645427). We carried out a systematic and comprehensive search on electronic databases such as PubMed and Hinari. In addition, Google Scholar and ScienceDirect were utilized to find relevant studies related to precancerous cervical lesions. Data from the included studies were extracted using an Excel spreadsheet and analyzed using STATA version 17. The methodological quality of the eligible studies was examined using the Joanna Briggs Institute (JBI) assessment

**Data availability statement:** All relevant data are within the manuscript and its Supporting Information files.

**Funding:** The author(s) received no specific funding for this work.

**Competing interests:** The authors have declared that no competing interests exist.

**Abbreviations:** ART, Antiretroviral therapy; AOR, Adjusted odds ratio; CIN, Cervical intraepithelial neoplasia; CI, Confidence Interval; CD4, Cluster of differentiation 4; DNA, Deoxyribonucleic acid; HIV, Human Immunodeficiency Virus; HPV, Human Papillomavirus; OCP, Oral contraceptive pills; PCL, Precancerous cervical lesions; STI, Sexual transmitted infections; VIA, Visual inspection with acetic acid; WHO, World health organization.

tool. Publication bias was checked by using the funnel plot and Egger's tests. A random-effects model using the Der Simonian Laird method was used to estimate the pooled prevalence of pre-cancerous cervical lesions in Africa. The I-squared and Cochrane Q statistics were used to assess the level of statistical heterogeneity among the included studies.

## Results

A total of 112 eligible articles conducted in Africa, encompassing 212,984 study participants, were included in the quantitative meta-analysis. Thus, the pooled prevalence of pre-cancerous cervical lesions in Africa was 17.06% (95% confidence interval: 15.47%−18.68%). In this review, having no formal education (AOR = 4.07, 95% CI: 1.74, 9.53), being rural dweller(AOR = 2.38, 95% CI: 1.64, 3.46), history of STIs (AOR = 3.94, 95% CI: 2.97, 5.23), history of having multiple partners (AOR = 2.73, 95% CI: 2.28, 3.28), early initiation of coitus (AOR = 2.77, 95% CI: 2.11, 3.62), being HIV-seropositive women (AOR = 3.33, 95% CI: 2.32, 4.78), a CD4 count <200 cells/mm³ (AOR = 5.17, 95% CI: 1.70, 15.71), not being on ART (AOR = 2.58, 95% CI: 1.45, 4.58), smoking (AOR = 3.91, 95% CI: 1.43, 10.67) and prolonged use of oral contraceptive pills (AOR = 4.39, 95% CI: 2.77, 6.96) were significantly associated with precancerous cervical lesions.

## Conclusions

In Africa, the overall prevalence of pre-cancerous cervical lesions is high (17%). The findings of this review highlight that health professionals, health administrators, and all other concerned bodies need to work in collaboration to expand comprehensive cervical cancer screening methods in healthcare facilities for early detection and treatment of cervical lesions. In addition, increasing community awareness and health education, expanding visual inspection of the cervix with acetic acid in rural areas, offering special attention to high-risk groups (HIV-positive women), encouraging adherence to antiretroviral therapy for HIV-positive women, overcoming risky sexual behaviors and practices, and advocating early detection and treatment of precancerous cervical lesions.

## Introduction

Cervical cancer is the fourth most prevalent malignancy and the fourth leading cause of mortality in women globally [1]. In 2022, it is responsible for approximately 662,000 new cases and around 349,000 deaths [2]. It is the most common cancer among women in 25 countries, many of which are in Sub-Saharan Africa [2]. Despite being a global public health concern, cervical cancer remains the second most prevalent malignancy and the second leading cause of cancer-related death among women in low- and middle-income countries, including in Africa [1,3].

A precancerous cervical lesion (PCL), or cervical intraepithelial neoplasia (CIN), also known as cervical dysplasia, refers to abnormal cellular changes occurring in the transformation zone of the cervix [2,4]. Histologically, it's commonly classified into three grades, such as CIN 1 (mild), CIN 2 (moderate), and CIN 3 (severe). Cervical carcinogenesis typically begins with the formation of CIN, which progresses from mild dysplasia (CIN1) to high-grade squamous intraepithelial lesions (CIN2, CIN3), which is thought to be a true precursor to advanced cervical cancer [5].

Human papillomavirus (HPV) infection causes nearly 99.7% of precancerous and malignant cervical lesions [6–8]. Human Papillomavirus (HPV) infections are typically transient, causing mild and self-limiting lesions. However, when the infection persists, it can progress to PCL and cervical cancer [9–11]. Persistent HPV infection plays a critical role in the cause and progression of CIN, particularly high-risk types such as HPV 16 and 18, which together are responsible for over 70% of cervical cancer cases [2]. Persistent high-risk HPV strains play a pivotal role in the disease's pathogenesis by integrating their DNA into host cells and disrupting normal cell cycle regulation [12].

In sub-Saharan Africa, cervical cancer is a significant cause of cancer-related morbidity and mortality among women, primarily due to chronic HPV infections and the prevalent occurrence of HIV co-infection [13]. Nevertheless, antiretroviral therapy (ART) has been shown to decrease the likelihood of HPV infection, enhance the body's capacity to eradicate the virus, and decrease the risk of developing precancerous cervical lesions and its progression into cervical cancer [14,15]. Early detection and treatment of precancerous lesions often halt the gradual progression into cervical uterine cancer [16]. Progression to invasive cervical cancer can take more than 20 years after initial HPV infection, with CIN1 developing slowly and potentially progressing rapidly to high-grade lesions (CIN2 or CIN3) [17,18].

The WHO has launched a Global Strategy to Accelerate Cervical Cancer Elimination by 2030 that involves three important strategies: vaccination against HPV, screening, and treatment [2,19]. In 2020, the WHO introduced the 90-70-90 global plan for preventing cervical cancer. This plan targets 90% of girls fully vaccinated against HPV by age 15 years, 70% of women to be screened for cervical disease with a high-performance test at least twice by the age of 45, and 90% of women with PCL and invasive cervical cancer should receive appropriate treatment [2,20,21].

The primary strategy to combat cervical cancer is HPV vaccination, which effectively prevents the development of precancerous lesions that increase the risk of cervical cancer in women [22]. Moreover, regular and timely cervical cancer screening methods, including Pap smear tests, HPV DNA testing, dual-stain cytology, and visual inspection, serve as a crucial secondary prevention strategy that facilitates early detection and treatment of precancerous lesions, making cervical cancer largely preventable [2,18,23]. However, the majority of women in countries with limited resources have no access to early detection and treatment of PCL. Screening and diagnosis of PCL remains a major challenge in LMICs [24,25]. In resource-limited areas, visual inspection of the cervix with acetic acid (VIA) or with Lugol's iodine (VILI) followed by treatment (screen-and-treat approach) is the best alternative approach for secondary prevention of invasive cervical cancer [26,27].

Since cervical cancer is the leading cause of cancer-related death for women globally, proper CIN treatment approaches are essential [28]. Effective cervical cancer screening programs, coupled with timely treatment of abnormal findings, can reduce the disease burden by up to 80% [2,29]. Both excisional techniques (cold knife conization, loop electrosurgical excision procedure, or LEEP) and ablative techniques (laser ablation, thermal ablation, and cervical cryotherapy) are effective therapeutic methods for PCL [30,31]. In prior studies women's age, educational status, history of STIs, history of multiple partners, early initiation of sexual intercourse, being HIV-positive, smoking, and prolonged use of oral contraceptive pills were significantly associated with precancerous cervical lesions [13,32–34].

Cervical cancer is notably preventable and manageable through early detection at the pre-invasive stage, widespread HPV vaccination before sexual debut, and access to effective treatment [2]. These measures can significantly reduce individual risk and the broader burden of morbidity and mortality [2,35]. Despite the high morbidity and mortality associated with PCL and cervical cancer in Africa, HPV vaccination coverage (41.38%) and cervical cancer screening uptake (21%) remain significantly below the WHO's 90–70–90 global targets [36,37]. Furthermore, existing systematic reviews

in the region are limited and often characterized by methodological inconsistencies and variations in study populations [13,38,39].

Most existing evidence primarily focuses on the incidence and mortality of invasive cervical cancer, HPV vaccination coverage, or overall screening uptake. However, there is a scarcity of consolidated evidence on PCL and its associated factors in Africa. This systematic review and meta-analysis aimed to synthesize evidence from multiple studies to estimate the pooled prevalence of PCL and identifying determinants among women who underwent cervical cancer screening, irrespective of age or HIV status. This comprehensive evidence is crucial for evaluating the effectiveness of screening programs and implementing evidence-based interventions, specifically as PCL represent a key target for timely treatment and prevention of invasive cervical cancer. It provides evidence-based insights to strengthen both facility-based and community-based cervical cancer screening approaches. Ultimately, it enables the detection and treatment of precancerous cervical lesions, improves survival rates, and reduces morbidity and mortality from cervical cancer.

## Research questions

1. What is the pooled prevalence of precancerous cervical lesions among women screened for cervical cancer in Africa?

2. What are the determinant factors of precancerous cervical lesions among women screened for cervical cancer in Africa?

## Materials and methods

### Study protocol and search strategy

This systematic review followed the Preferred Reporting Items for Systematic Reviews and Meta-Analysis (PRISMA) guidelines [40] (S1 file). The study protocol was developed and registered on PROSPERO (ID: CRD42025645427).

We conducted a comprehensive search on Google Scholar, PubMed, Hinari, and ScienceDirect for primary studies of precancerous cervical lesions and its determinant factors undertaken in Africa. The search technique was based on the condition, context, and study population (CoCo Pop) framework [41]. Publications were retrieved from prior studies meeting the eligibility criteria. A search strategy was developed for databases by combining keywords using Boolean operators. Both published and unpublished articles between January 1, 2015, and February 20, 2025, were included. Finally, we used the following combination of searching terms: "prevalence," "magnitude," "burden," "precancerous cervical lesions," "premalignant cervical lesions," "cervical intraepithelial neoplasia," "CIN", "cervical lesions," Africa. In addition, snowballing techniques were used to retrieve further studies from the citation list of papers identified in the available databases (S2 file).

### Eligibility criteria

**Inclusion criteria.** Condition: The condition of interest was pre-cancerous cervical lesions among women who underwent cervical cancer screening.

Context: All primary studies that reported the prevalence and/or associated factors of precancerous cervical lesions in the Africa context.

Population: The study population included all women aged 15 years and above, regardless of HIV status, who underwent cervical cancer screening.

Study design: All primary observational studies, including cross-sectional, case-control, and cohort studies that reported the prevalence and/or associated factors of pre-cancerous cervical lesions in Africa.
Publication year: Studies published between 2015 and 2025.

**Exclusion criteria.** Articles were excluded for the following reasons: the article did not report the outcome of interest, narrative reviews, qualitative reviews, expert opinions, case reports, editorials, correspondence, abstracts, and methodological studies.

## Measurement of outcome variables

The primary objective of this study was to determine the overall prevalence of precancerous cervical lesions among women screened for cervical cancer in Africa. The magnitude of PCL was calculated by dividing the number of women who screened positive for precancerous cervical lesions by the total number of women in the study and then multiplying by 100. The second objective of this study was to identify factors associated with pre-cancerous cervical lesions in Africa, which were evaluated using adjusted odds ratios from prior studies.

The WHO recommends screening and treating precancerous lesions to prevent cervical carcinoma [35]. Cervical cancer screening is the practice of checking women for precancerous or malignant cells on the cervix using diagnostic procedures including the cytology (Pap smear), the HPV DNA test, or VIA [35,42]. These procedures involve taking cervical cells for microscopic analysis or diagnosing the presence of HPV, a virus associated with cervical cancer [42,43]. Histologically, CIN can be classified into three stages, CIN 1, CIN 2, and CIN 3. It can also divided into two stages: low-grade squamous intraepithelial lesions (LSIL, CIN 1) and high-grade squamous intraepithelial lesions (HSIL, CIN 2, 3) [2,5].

Additionally, a positive finding is an acetowhite lesion with well-defined margins near the transformation zone, or a white cervix (visual inspection with acetic acid). Visual inspection with acetic acid yields a negative test if there is no acetowhite lesion, but a visible ulcer with seeping and bleeding may indicate cancer [35]. This review included all studies that reported positive findings for PCL, regardless of participants' age, HIV status, or the cervical cancer screening methods employed.

## Quality assessment for included studies

The Joanna Briggs Institute (JBI) Critical Appraisal Checklist for cross-sectional studies was used to evaluate the quality of included articles in this study [41]. This quality assessment checklist contains nine items, ranging from 0 to 9. Studies that scored 5 or higher on the JBI checklist were considered high quality and therefore included in the review. The studies' quality was evaluated independently by two authors, BAM and NTT. Any disputes in quality assessment between these two authors were handled through open discussion and consultation with a second author, GMA.

## Data extraction and management

Two authors (BAM and NTT) carried out data extraction from the included articles using a standardized data abstraction form developed in an Excel spreadsheet. Based on the inclusion and exclusion criteria, all searched studies were transferred to EndNote 20, reference management software [44]. Articles were screened and selected first based on their title and abstract, and then the full text was reviewed. In cases of dispute, discussions with third reviewers (GMA) were held to determine the final article selection to include in this review. Following the comprehensive searching, possibly eligible publications were imported into Endnote software. Duplicate studies were deleted in cases where two or more papers shared similar features. Structured data extraction in a Microsoft Excel spreadsheet was designed and implemented. For each primary study, the following data were extracted: identification data (first author's last name and publication year), prevalence of PCL, factors associated with PCL, adjusted odds ratio with 95% confidence intervals, study area, sample size, publication bias assessment methods, risk bias assessment method, and scores (S3 file).

## Data synthesis and statistical analysis

Data were extracted using a Microsoft Excel spreadsheet and then exported to STATA 17 statistical software, where all statistical data analyses were performed [45]. The extracted data were presented as texts, tables, and forest plots. The standard error of prevalence for each study was calculated using a binomial distribution. The pooled prevalence of the studies was examined for heterogeneity using the Higgins I-squared ($I^2$) test. Heterogeneity among those included was characterized as low, moderate, or high based on I-square values of <25%, 50%−75%, and 75%, respectively [46].

A random-effects meta-analysis approach (Der Simonian and Laird's method) was employed to estimate the pooled prevalence of PCL in Africa. Subgroup analysis was performed across the country, regions of Africa, screening methods, and study population to identify potential sources of study heterogeneity. Additionally, we conducted a leave-one-out sensitivity analysis to examine the effect of individual studies on the pooled estimate. The pooled estimates across the continent were then displayed using forest plots and tables, along with their respective 95% confidence intervals. Graphically, publication bias was examined using a forest plot (23). Furthermore, the statistical significance of publication bias was tested using both Egger's and Begg's tests, and a p-value less than 0.05 was employed to confirm the existence of publication bias [47]. A trim and fill analysis could be performed to determine the number of potentially missed studies and estimate the adjusted pooled prevalence of PCL. In the end, the pooled adjusted odds ratio (AOR) with 95% confidence intervals was displayed using forest plots.

## Results

### Study selection and characteristics of the included studies

A total of 1,187 studies were retrieved through all searching databases, including Google Scholar; 312 duplicate records were removed, and further screening was performed for the remaining 875 studies. However, we excluded the majority of studies (n = 736) by reading their titles and abstracts. Then, the remaining 139 full-text articles were examined for eligibility criteria, and 27 studies were dropped for different reasons, such as variation in the study context, insufficient data, not being directly related to the outcome of interest, scoping reviews, and qualitative reviews. In total, 112 eligible articles were included in the final quantitative meta-analysis [48–144] (Fig 1).

This systematic review and meta-analysis included a total of 212,984 women who underwent cervical cancer screening. In terms of the distribution of the studies across Africa, 54 and 41 studies were conducted in East Africa and West Africa, respectively (Table 1, S3 file).

### Magnitude of precancerous cervical lesions in Africa

In this study, the pooled prevalence of precancerous cervical lesions in Africa was 17.06% (95% confidence interval: 15.47%−18.68%). The statistical test found significant heterogeneity among the included studies (heterogeneity $I^2$ = 99.19%, p-value = 0.000). Thus, a random-effects meta-analysis model was applied.

### Heterogeneity and sub-group analysis

A subgroup analysis was carried out based on the country in which the study was conducted, the study population, and cervical cancer screening methods. Accordingly, the highest prevalence was found in Zambia, 42.79% (95% CI: 3.89, 89.46), whereas the lowest prevalence was detected in Mali, 6.72% (95% CI: 1.82, 11.62). In addition, a subgroup analysis was carried out based on the study population; a higher overall magnitude of PCL was found among women with HIV (22.87%; 95% CI: 17.28, 28.46). Regarding the screening methods, the highest overall prevalence of PCL was found in HPV DNA and colposcopy diagnostic methods, 24.92% (95% CI: 4.35, 45.50). While the lowest overall prevalence was reported in VIA screening methods, 14.60% (95% CI: 13.13, 16.08). Despite conducting subgroup analysis based on the aforementioned factors, no significant improvement in heterogeneity in the pooled estimate of PCL (Table 2).

### Publication bias, trim and fill analysis

A funnel plot was used to visually check the presence of publication bias, while Egger's test was employed to confirm it. In this study, the funnel plot looks slightly asymmetrical, which shows the existence of publication bias among the studies. Statistically, Egger's (p-value = 0.000) and Begg's tests (p-value = 0.000) were statistically significant, indicating the presence of publication bias (Fig. 2).

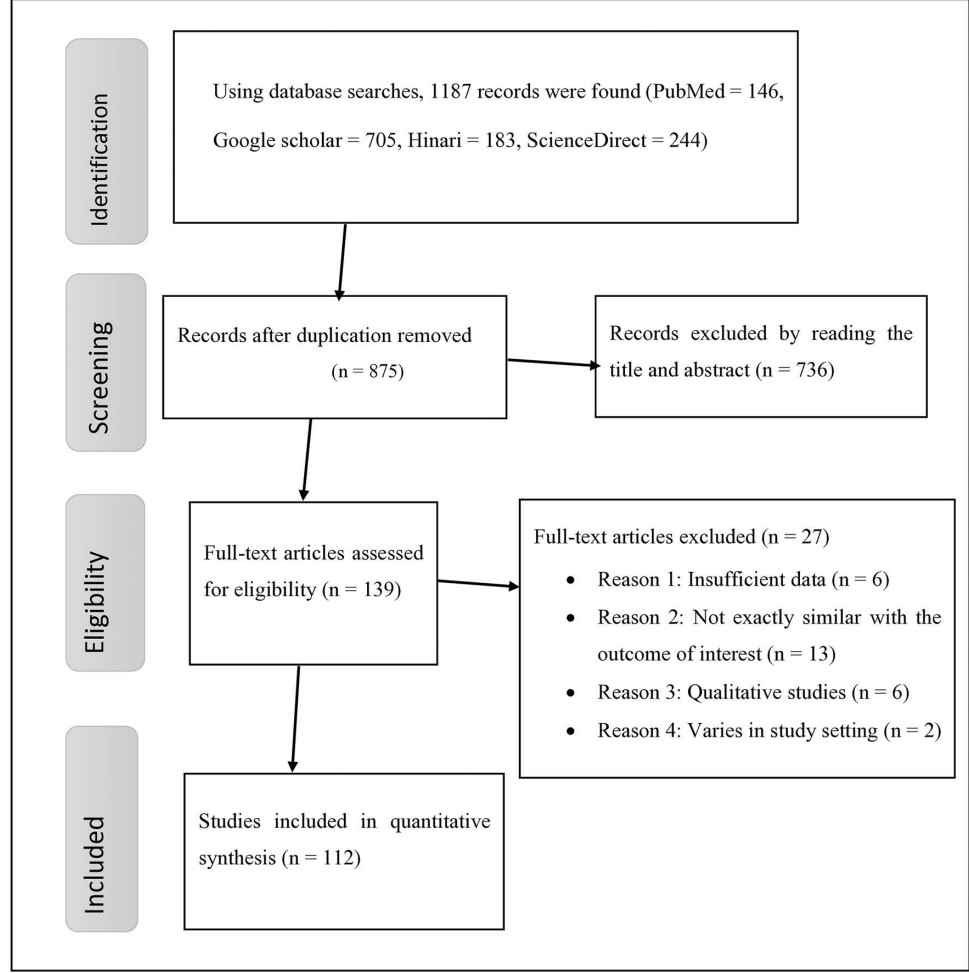

**Fig 1. Prisma flow diagram showing the selection of studies for precancerous cervical lesions in Africa.**

A non-parametric trim and fill statistical analysis was performed to determine the number of potentially missing studies to reduce and adjust for publication bias in the included studies. However, the trim and fill analysis revealed the absence of significant publication bias because the overall prevalence of observed studies was equal to the sum of observed and imputed studies (Table 3).

### Sensitivity analysis

A leave-one-out sensitivity analysis using the random-effects model was performed to examine the effect of a single study on the estimated effect size. However, the findings show that a single study did not significantly affect the total effect size, and the point estimate of the omitted study falls within the confidence interval of the overall estimate of PCL. This proved the reliability of the pooled estimate of precancerous cervical lesions in Africa (S4 file).

### Factors associated with precancerous cervical lesions in Africa

This systematic review and meta-analysis examined 24 publications that reported factors associated with precancerous cervical lesions in Africa. In this study, no formal education, rural residency, history of STIs, history of multiple partners,

**Table 1. Descriptive summary of studies included in systematic review of the prevalence of precancerous cervical lesions (PCL) in Africa.**

| Authors | Study design | Study area | Study population | Diagnosis method | Mean Age | Sampe size | PCL (%) | JBI score |
|---|---|---|---|---|---|---|---|---|
| Abera et al, 2021 [48] | CS | Ethiopia | >21 years | VIA | 35.69 | 883 | 8.95 | 9 |
| Adam et al, 2022 [49] | Case-control | Sudan | >21 years | Pap smear | NR | 109 | 47.6 | 7 |
| Adam et al, 2022 [49] | Case-control | Sudan | >21 years | Pap smear | NR | 109 | 4.6 | 8 |
| Ago et al., 2016 [50] | CS | Nigeria | >21 years | Pap smear | 31.29 | 100 | 3 | 7 |
| Ali et al, 2019 [51] | CS | Ethiopia | >21 years | Pap smear | 42.7 | 366 | 13.1 | 9 |
| Ambounda-Ledaga et al., 2024 [52] | Cohort | Gabon | HIV positive | VIA | 41 | 115 | 28.7 | 8 |
| Awolude et al., 2021 [53] | Cohort | Nigeria | HIV positive | Pap smear | 37.1 | 468 | 10 | 8 |
| Bateman et al, 2015 [54] | CS | Zambia | HIV positive | VIA | NR | 309 | 66.63 | 9 |
| Belayneh et al, 2019 [55] | CS | Ethiopia | HIV positive | Pap smear | 34.48 | 284 | 9.9 | 9 |
| Chibvongodze et al, 2017 [56] | CS | Kenya | >21 years | Pap smear | NR | 25 | 12 | 9 |
| Chris-Ozoko et al., 2020 [57] | CS | Nigeria | >21 years | Pap smear | NR | 2115 | 7 | 9 |
| Christensen et al, 2023 [145] | CS | Uganda | >21 years | VIA | NR | 3946 | 7.6 | 9 |
| Daniel et al., 2019 [58] | CS | Nigeria | HIV positive | Pap smear | 41.2 | 326 | 12.2 | 8 |
| Darré et al, 2024 [59] | CS | Togo | HIV positive | Pap smear | 47 | 271 | 11.4 | 7 |
| Eseoghene, et al 2021 [69] | CS | Nigeria | HIV positive | Pap smear | 32.4 | 342 | 16.67 | 7 |
| Deksissa et al, 2015 [60] | CS | Ethiopia | >21 years | VIA | 46.5 | 334 | 12.9 | 9 |
| Derbie et al, 2022 [61] | NR | Ethiopia | >21 years | VIA | NR | 335 | 19.1 | 8 |
| Desire et al, 2016 [62] | CS | DRC | >21 years | VIA | 35.22 | 229 | 38 | 9 |
| Ephrem Dibisa et al, 2022 [68] | CS | Ethiopia | >21 years | VIA | 30 | 399 | 27.4 | 9 |
| Diop et al, 2022 [63] | CS | Senegal | >21 years | VIA | NR | 385 | 5.45 | 8 |
| Doh et al, 2021 [64] | CS | Cameroon | >21 years | Pap smear | 42.3 | 482 | 26.6 | 9 |
| Donkhon et al., 2019 [65] | CS | Ghana | >21 years | Pap smear | NR | 592 | 3.7 | 8 |
| Effah et al., 2024 [66] | CS | Ghana | HIV positive | Colposcopy | NR | 258 | 8.5 | 8 |
| Effah et al., 2024 [66] | Cohort | Ghana | >21 years | Colposcopy | 38.8 | 100 | 11.4 | 7 |
| Eljabuet al, 2021 [67] | CS | Lybia | >21 years | Pap smear | 39.54 | 507 | 4 | 8 |
| Essmat et al, 2021 [70] | CS | Egypt | >21 years | Pap smear | 34.18 | 830 | 17 | 8 |
| Fentie et al, 2020 [71] | CS | Ethiopia | >21 years | VIA | 35.74 | 844 | 10.3 | 7 |
| Getinet et al, 2021 [73] | CS | Ethiopia | >21 years | VIA | 37.11 | 340 | 15.3 | 9 |
| Getinet et al, 2024 [72] | CS | Ethiopia | >21 years | Pap smear | 35.02 | 391 | 14 | 9 |
| Getinet et al, 2024 [72] | CS | Ethiopia | >21 years | VIA | 36.7 | 337 | 13.9 | 9 |
| Gnatou et al, 2023 [74] | CS | Togo | >21 years | VIA | 36* | 728 | 3.9 | 8 |
| Hailemariam et al, 2017 [76] | CS | Ethiopia | >21 years | VIA | 41.66 | 2120 | 16.5 | 9 |
| Hailemariam et al, 2020 [75] | CS | Ethiopia | >21 years | VIA | 32.8 | 412 | 9 | 8 |
| Hayumbu al, 2021 [77] | CS | Zambia | >21 years | VIA | 37* | 329 | 19 | 8 |
| Hoffman et al, 2016 [78] | CS | South Africa | >21 years | Pap smear | 50.5 | 237 | 31.7 | 9 |
| Ibrahima et al., 2023 [79] | Cohort | Guinea | >21 years | VIA | 36 | 2308 | 9.6 | 9 |
| Inuwa et al., 2016 [80] | CS | Nigeria | HIV positive | Pap smear | 30.58 | 365 | 47.94 | 9 |
| Inuwa et al., 2016 [80] | CS | Nigeria | HIV negative | Pap smear | 30.58 | 365 | 31.78 | 9 |
| Irabor et al., 2018 [81] | CS | Nigeria | >21 years | Pap smear | 43 | 525 | 17 | 9 |
| Jolly et al, 2017 [144] | CS | Swaziland | HIV positive | VIA | NR | 273 | 22.9 | 7 |
| Jolly et al, 2017 [144] | CS | Swaziland | HIV negative | VIA | NR | 273 | 5.7 | 9 |
| Kagoné et al., 2022 [82] | CS | Burkina Faso | >21 years | VIA | 34.9 | 577 | 15.4 | 7 |
| Kamdem et al, 2022 [83] | CS | Cameroon | >21 years | Colposcopy | 44 | 71 | 57.5 | 9 |
| Karuri et al, 2015 [84] | CS | Kenya | >21 years | Pap smear | 39.9 | 119 | 36 | 8 |
| Kaseka et al, 2022 [85] | Cohort | Malawi | >21 years | Pap smear | 41.99 | 500 | 24.4 | 7 |

*(Continued)*

**Table 1.** (Continued)

| Authors | Study design | Study area | Study population | Diagnosis method | Mean Age | Sampe size | PCL (%) | JBI score |
|---|---|---|---|---|---|---|---|---|
| Kassa LS et al, 2019 [86] | CS | Ethiopia | HIV positive | VIA | 35.9 | 435 | 20.2 | 8 |
| Katz et al., 2016 [87] | Cohort | South Africa | HIV positive | Pap smear | 33* | 237 | 75.6 | 8 |
| Kirabira et al, 2024 [88] | CS | Uganda | HIV positive | Pap smear | 45 | 210 | 23 | 7 |
| Kirabira et al, 2024 [88] | CS | Uganda | HIV negative | Pap smear | 45 | 210 | 12 | 7 |
| Kiros et al, 2021 [89] | CS | Ethiopia | HIV positive | VIA | 36.96 | 427 | 9.3 | 8 |
| Kiros et al, 2021 [89] | CS | Ethiopia | HIV negative | VIA | 36.96 | 129 | 8.6 | 9 |
| Kurtay et al, 2022 [90] | Cohort | Somalia | >21 years | Pap smear | NR | 497 | 12.3 | 9 |
| Lawal et al., 2017 [91] | CS | Nigeria | HIV positive | Pap smear | 34.5 | 135 | 56.3 | 9 |
| Lawal et al., 2017 [91] | CS | Nigeria | HIV negative | Pap smear | 34.5 | 135 | 12.6 | 8 |
| Ledaga et al., 2022 [92] | CS | Gabon | >21 years | Pap smear | 34.3 | 144 | 11.8 | 9 |
| Lemma et al., 2024 [93] | CS | Ethiopia | HIV positive | VIA | 35.05 | 257 | 16 | 7 |
| Lemu et al, 2021 [94] | CS | Ethiopia | HIV positive | VIA | 34 | 444 | 18.7 | 9 |
| Macharia et al, 2017 [95] | CS | Kenya | HIV positive | VIA | 38.3 | 75 | 22.7 | 8 |
| Magaji et al, 2024 [96] | CS | Nigeria | HIV positive | Pap smear | 45.08 | 566 | 24 | 9 |
| Makuza et al, 2015 [97] | CS | Uganda | >21 years | VIA | 37 | 1002 | 1.7 | 8 |
| Mariko et al, 2021 [98] | CS | Mali | >21 years | VIA | 38.66 | 2351 | 4.2 | 7 |
| Mayeri et al., 2024 [99] | CS | DRC | >21 years | Pap smear | 37.5 | 142 | 17 | 7 |
| Mekuria et al, 2021 [100] | CS | Ethiopia | >21 years | VIA | NR | 422 | 23.5 | 9 |
| Merera et al, 2021 [101] | CS | Ethiopia | >21 years | VIA | 35.97 | 293 | 15.7 | 9 |
| Misgina et al, 2016 [102] | CS | Ethiopia | >21 years | VIA | 32.95 | 342 | 6.9 | 8 |
| Mremi et al, 2022 [103] | CS | Tanzania | >21 years | VIA | NR | 1620 | 17.6 | 9 |
| Muia et al., 2021 [104] | NR | Kenya | HIV positive | Pap smear | NR | 385 | 7.8 | 8 |
| Mukanyangezi et al, 2018 [105] | Cohort | Rwanda | HIV positive | Pap smear | 46 | 400 | 24.3 | 9 |
| Mukanyangezi et al, 2018 [105] | Cohort | Rwanda | HIV negative | Pap smear | 46 | 400 | 9.9 | 7 |
| Mulugeta Y., 2022 [106] | CS | Ethiopia | HIV positive | VIA | 34.97 | 334 | 16.5 | 8 |
| Mulugeta Y., 2022 [106] | CS | Ethiopia | HIV negative | VIA | 34.97 | 363 | 9.6 | 8 |
| Musa et al., 2020 [107] | CS | Nigeria | HIV positive | Pap smear | 39 | 1556 | 3.7 | 8 |
| MUTUKU et al, 2024 [108] | CS | Kenya | HIV positive | Pap smear | NR | 400 | 7.8 | 8 |
| Ngwibete et al., 2024 [109] | CS | Nigeria | >21 years | VIA | 31 | 148 | 10.8 | 7 |
| Njagi et al, 2021 [110] | CS | Kenya | HIV positive | Pap smear | 30.2 | 373 | 42 | 8 |
| Njagi et al, 2021 [110] | CS | Kenya | HIV negative | Pap smear | 30.2 | 107 | 19 | 8 |
| Nkfusai et al, 2017 [111] | CS | Cameroon | >21 years | VIA | 38.5 | 60 | 3.33 | 7 |
| Nzang et al., 2024 [113] | Cohort | Cameroon | >21 years | VIA | 39.8 | 224 | 13.4 | 9 |
| Oduor et al., 2018 [114] | Cohort | Kenya | >21 years | Pap smear | 39.8 | 425 | 53.4 | 8 |
| Okorie et al., 2017 [115] | CS | Nigeria | HIV positive | Pap smear | 41.2 | 226 | 10.6 | 8 |
| OKUNADE et al., 2023 [116] | CS | Nigeria | >21 years | Pap smear | 38.6 | 593 | 6.7 | 9 |
| Okunowo et al., 2023 [117] | CS | Nigeria | >21 years | Pap smear | 35.63 | 244 | 9.8 | 7 |
| Okwi et al, 2017 [118] | CS | Uganda | >21 years | Pap smear | 43.3 | 1210 | 11.2 | 9 |
| Omeke et al., 2022 [119] | CS | Nigeria | >21 years | Pap smear | 42.7 | 212 | 15 | 7 |
| Omoragbon et al., 2017 [143] | CS | Nigeria | HIV positive | Pap smear | 44 | 65 | 26.2 | 7 |
| Omoragbon et al., 2017 [143] | CS | Nigeria | HIV negative | Pap smear | 39.35 | 65 | 16.9 | 8 |
| Omoyeni et al, 2022 [120] | CS | South Africa | >21 years | Pap smear | 39.73 | 246 | 48.2 | 9 |
| Oringo J., 2020 [121] | CS | Uganda | HIV positive | Pap smear | 39.73 | 210 | 5.7 | 7 |
| Oumar et al, 2022 [134] | CS | Mali | >21 years | VIA | NR | 42492 | 9.2 | 8 |
| Paluku et al., 2019 [122] | CS | DRC | >21 years | Pap smear | NR | 644 | 7.45 | 9 |

*(Continued)*

**Table 1.** (Continued)

| Authors | Study design | Study area | Study population | Diagnosis method | Mean Age | Sampe size | PCL (%) | JBI score |
|---|---|---|---|---|---|---|---|---|
| Rantshabeng et al., 2024 [123] | CS | Botswana | >21 years | Pap smear | 32* | 171 | 13.5 | 9 |
| Ntuliet et al., 2020 [112] | Cohort | South Africa | HIV positive | Pap smear | 38.78 | 39220 | 31.8 | 9 |
| Ntuliet al., 2020 [112] | Cohort | South Africa | HIV negative | Pap smear | 41.5 | 45246 | 9.2 | 9 |
| Siad et al., 2023 [124] | CS | Somalia | >21 years | VIA | NR | 925 | 15.7 | 8 |
| Simo et al., 2021 [125] | CS | Cameroon | >21 years | Pap smear | NR | 189 | 17 | 9 |
| Simo et al., 2021 [125] | CS | Cameroon | >21 years | Pap smear | NR | 137 | 15.3 | 9 |
| Ssedyabane et al., 2024 [126] | CS | Uganda | >21 years | Pap smear | 41.9 | 351 | 6.6 | 8 |
| Stroetmann et al., 2024 [127] | CS | Ethiopia | >21 years | VIA | NR | 13800 | 6.9 | 7 |
| Teame et al., 2018 [128] | Case-control | Ethiopia | >21 years | VIA | NR | 343 | 12.8 | 9 |
| Temesgen et al., 2021 [130] | CS | Ethiopia | >21 years | VIA | 35* | 337 | 13.1 | 9 |
| Temesgen et al, 2019 [129] | CS | Ethiopia | >21 years | VIA | 36.26 | 422 | 6.9 | 9 |
| Temesgen et al, 2020 [131] | CS | Ethiopia | >21 years | VIA | 35* | 234 | 14.1 | 8 |
| Tenkir et al, 2023 [132] | CS | Ethiopia | >21 years | VIA | 38.68 | 372 | 14 | 8 |
| Tirkaso et al, 2024 [133] | CS | Ethiopia | >21 years | Pap smear | 47.06 | 2613 | 16.8 | 7 |
| Ugboaja et al., 2016 [135] | CS | Nigeria | HIV positive | Pap smear | 37.7 | 110 | 28.2 | 9 |
| Umemmuo MU et al., 2019 [136] | Cohort | Nigeria | >21 years | Pap smear | 51.2 | 5212 | 6.5 | 8 |
| Vieira et al., 2024 [137] | Cohort | Cape Vede | >21 years | Pap smear | 38* | 13035 | 16.9 | 9 |
| Wabo et al, 2022 [138] | CS | Cameroon | >21 years | Pap smear | 40.2 | 925 | 12.2 | 9 |
| Wakwoya et al, 2020 [139] | CS | Ethiopia | >21 years | VIA | 32.5 | 1181 | 24.5 | 9 |
| Worku et al, 2024 [140] | CS | Ethiopia | HIV positive | VIA | 36* | 915 | 24.48 | 9 |
| Woromogol et al, 2021 [141] | CS | Gabon | >21 years | VIA | 39.8 | 191 | 22.5 | 8 |
| Zelalem et al, 2022 [142] | CS | Ethiopia | HIV positive | VIA | 39.7 | 267 | 7.5 | 8 |

Key: *Median age of the woman, CS: Cross-sectional study, NR: Not reported, PCL: Precancerous cervical lesions, VIA: Visual inspection with acetic acid

early initiation of coitus, HIV seropositive women, low CD4 count, not being on ART, smoking, and OCP were all significantly associated with precancerous cervical lesions in Africa.

In this meta-analysis, women who had no formal education were four times more likely (AOR = 4.07, 95% CI: 1.74, 9.53) to develop PCL than those who had formal education. Women from rural areas were 2.38 times more likely (AOR: 2.38, 95% CI: 1.64, 3.458) to develop precancerous cervical lesions as compared to women living in urban areas. The odds of PCL were 3.94 times more likely in women with a history of STIs (AOR = 3.94, 95% CI: 2.97, 5.23) than in women who did not have a history of STIs.

According to the pooled odds ratio of 21 studies, it was found that having a history of multiple sexual partners was significantly associated with developing precancerous cervical lesions. Moreover, women with multiple sexual partners had 2.73 times (AOR = 2.73; 95% CI: 2.28, 3.28) higher likelihood of developing precancerous cervical lesions compared to their counterparts. Women who began sexual intercourse early, before 18 years, were three times more likely to develop PCL than women who did not begin sexual intercourse early. In addition, the odds of developing precancerous cervical lesions among HIV-positive women were 3.33 times (AOR = 3.33; 95% CI: 2.32, 4.78) higher than those of HIV-negative women.

In this meta-analysis, CD4 count was identified as an independent factor significantly associated with the occurrence of PCL. HIV-positive women with CD4 counts below 200 cells/mm³ had five times higher odds of developing PCL than those with CD4 counts 200 cells/mm³ or above. The odds of developing precancerous cervical lesions among women who did not start ART were 2.58 times higher than those who initiated ART (AOR: 2.58, 95% CI: 1.45, 4.580).

**Table 2. Sub-group analysis of precancerous cervical lesions among women who screened in Africa.**

| Parameters | Studies | Prevalence (%) at 95% CI | I² (%) | P-value |
|---|---|---|---|---|
| **Country** | | | | |
| South Africa | 4 | 39.08% (24.47, 53.69) | 99.95 | 0.000 |
| Rwanda | 2 | 17.02% (2.91, 31.13) | 96.71 | 0.000 |
| Somalia | 2 | 14.12% (10.78, 17.43) | 68.85 | 0.073 |
| Sudan | 2 | 25.88% (8.25, 68.02) | 98.54 | 0.000 |
| Swaziland | 2 | 14.17% (2.69, 31.02) | 97.15 | 0.000 |
| Cameroon | 6 | 19.5% (12.11, 26.89) | 94.95 | 0.000 |
| DRC | 3 | 20.65% (92.68, 38.62) | 97.68 | 0.000 |
| Gabon | 3 | 20.63% (10.97, 30.28) | 85.49 | 0.001 |
| Ghana | 3 | 7.21% (2.67, 11.75) | 81.59 | 0.004 |
| Kenya | 7 | 25.17% (11.80, 38.54) | 98.39 | 0.000 |
| Mali | 2 | 6.72% (1.82, 11.62) | 99.24 | 0.000 |
| Togo | 2 | 7.44% (0.10, 14.77) | 92.46 | 0.000 |
| Uganda | 6 | 9.11% (5.55, 12.68) | 96.94 | 0.000 |
| Zambia | 2 | 42.79% (3.89, 89.47) | 99.48 | 0.000 |
| Ethiopia | 30 | 14.20% (12.05, 16.35) | 96.32 | 0.000 |
| Nigeria | 19 | 17.06% (15.47, 18.64) | 97.35 | 0.000 |
| Others* | 8 | 13.67% (9.79, 17.55) | 97.81 | 0.000 |
| **Study population** | | | | |
| Age > 21 years | 69 | 14.63% (13.34, 15.91) | 97.83 | 0.000 |
| HIV negative women | 11 | 12.95% (9.64, 16.25) | 91.38 | 0.000 |
| HIV positive women | 32 | 22.87% (17.29, 28.46) | 99.35 | 0.000 |
| **Screening methods** | | | | |
| VIA | 46 | 14.60% (13.13, 16.08) | 97.66 | 0.000 |
| Cytology (Pap smear) | 60 | 18.77% (15.97, 21.57) | 99.19 | 0.000 |
| HPV DNA & Colposcopy | 3 | 24.92% (4.35, 45.50) | 96.89 | 0.000 |

* Botswana, Burkina Faso, Cape Vede, Guinea, Lybia, Malawi

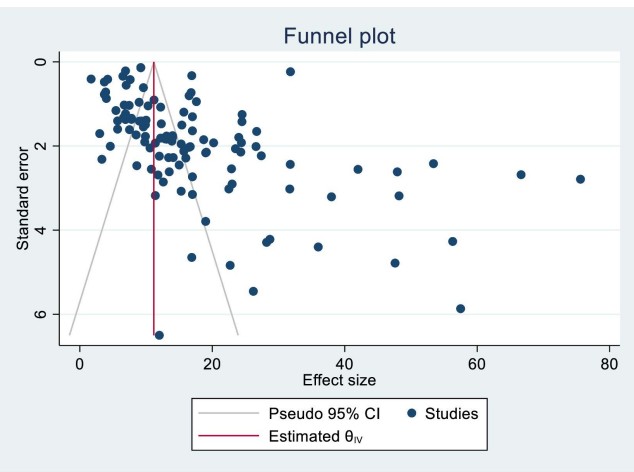

**Fig 2. A funnel plot test that demonstrating the prevalence of precancerous cervical lesions in Africa.**

**Table 3. Non-parametric trim and fill analysis of publication bias for precancerous cervical lesions among women who screened in Africa.**

| Studies | Effect size [95% conf. interval] | | |
|---|---|---|---|
| Observed | 17.06 | 15.47 | 18.68 |
| Observed + Imputed | 17.06 | 15.47 | 18.68 |

Concerning personal habits, those women who smoked cigarettes were four times more likely (AOR = 3.91, 95% CI: 1.43, 10.67) to develop PCL than those who did not smoke cigarettes. The odds of developing precancerous cervical lesions were 4.39 times more likely among prolonged OCP users (AOR = 4.39, 95% CI: 2.77, 6.96) when compared with those who were non-users (Table 4).

## Discussion

The main objectives of this systematic review and meta-analysis were to determine the magnitude of PCL and its associated factors in Africa. Thus, the overall prevalence of precancerous cervical lesions in Africa was 17.06% (95% CI: 15.47%–18.68%). This finding was slightly higher than studies conducted in Ethiopia that reported 15% [146,147]. Similarly, the magnitude of PCL in this study was higher than studies conducted in Ethiopia at 13% and in Latin America at 4% [34]. This difference could be due to variations in study population, sociocultural setting, and diagnostic or screening methods. The study population for the current study was all women screened for cervical cancer. While the study population for the comparable studies was women living with HIV (WLH). The lower prevalence in Latin America among HIV-positive women could be related to the expanded program of ART (72%) and the decreased number of new HIV infections by 14%, which collectively could reduce the incidence of PCL in the region [34,148]. However, there is a significantly higher HIV pandemic and AIDS-related death rate, particularly in Eastern and Southern Africa. Women and girls were approximately 58% of the total HIV infections, despite persistent gender inequities and widespread violence against females [148].

In contrast, the finding of this study was lower than the magnitude of PCL among HIV-positive women in Sub-Saharan Africa that was reported at 25.6% [13]. This was due to variations in the study population; a higher burden of PCL was found among women living with HIV in SSA [13]. There is evidence that the occurrence of precancerous cervical lesions and cervical cancer is strongly correlated with HIV infection [147,149]. Therefore, high-risk women, including HIV-positive women should undergo regular screening for cervical cancer with the available methods for early detection and treatment of PCL.

These findings underline the necessity for enhanced and comprehensive cervical cancer prevention interventions in Africa. Policymakers, health professionals, and public health systems should prioritize expanding primary prevention

**Table 4. Factors associated with precancerous cervical lesions among women who screened in Africa.**

| Variables | No_ studies | OR (95% CI) | I² (%) | P-value |
|---|---|---|---|---|
| No formal education | 2 | 4.07 (1.74, 9.53) | 92.3% | 0.001 |
| Rural residency | 2 | 2.38 (1.64, 3.46) | 53.2% | 0.000 |
| History of STIs | 24 | 3.94 (2.97, 5.23) | 88.3% | 0.000 |
| History of multiple partner | 21 | 2.73 (2.28, 3.28) | 27.7% | 0.000 |
| Early initiation of coitus | 18 | 2.77 (2.11, 3.62) | 70.9% | 0.000 |
| Seropositive women | 10 | 3.33 (2.32, 4.78) | 49.2% | 0.039 |
| Low CD4 count (<200cells/mm) | 3 | 5.17 (1.70, 15.71) | 82.9% | 0.003 |
| Not on ART | 3 | 2.58 (1.45, 4.58) | 0.0% | 0.001 |
| Smoking | 3 | 3.91 (1.43, 10.67) | 41.1% | 0.008 |
| Oral contraceptive pills | 7 | 4.39 (2.77, 6.96) | 24.8% | 0.000 |

through HPV vaccination (9–14 years) and strengthening secondary prevention by implementing accessible and population-based screening programs (e.g., VIA, cytological or HPV DNA testing). These initiatives contribute to advancing the WHO's global agenda for cervical cancer elimination and improving women's health outcomes across African countries.

With varying degrees of accuracy, this review combined the PCL magnitude using various cervical cancer screening techniques. Therefore, a subgroup analysis based on screening methods was conducted; the lowest overall prevalence of PCL was recorded in VIA screening methods (14.60%), whereas the highest pooled prevalence was identified in HPV DNA and colposcopy testing methods (24.92%). Nowadays, there are two methods for PCL screening and treatment. The initial approach is the screen-and-treat strategy, in which a positive primary screening test is the sole basis for treatment decisions [2]. The second technique is the screen, triage, and treat approach, in which the decision to treat relies on a positive initial screening test followed by a positive subsequent test, known as a "triage" test, with or without histological confirmation of diagnosis [2]. The World Health Organization (WHO) promotes a screen, triage, and treat approach with HPV DNA as the primary screening test for the general female population, followed by additional investigations [2]. However, visual examination with acetic acid (VIA) is an easily and highly cost-effective screening approach for premenopausal women in low- and middle-income countries (LMICs) [150–152].

In this systematic review and meta-analysis, precancerous cervical lesions are significantly more prevalent among HIV-positive women (22.87%) than HIV-negative women (12.95%). This finding was in agreement with studies done in Ethiopia and SSA [13,147,149]. This finding was in agreement with studies done in Ethiopia and SSA [34,147,149]. This could be explained by the fact that HIV-positive women are more susceptible to developing PCL due to their compromised immune system, which allows for chronic HPV infections, which is the leading cause of PCL and cervical cancer. Furthermore, co-infection with other STIs and having more lifetime sexual partners increase the risk. Therefore, integrating cervical cancer screening into ART clinics can enhance early detection and timely treatment of PCL, thereby reducing the risk of progression to invasive cervical cancer and improving health outcomes among these high-risk groups.

This study found that having no formal education, being a rural resident, having a history of STIs, having a history of multiple partners, early initiation of coitus, being an HIV-seropositive woman, having a low CD4 count, not being on ART, smoking, and prolonged use of OCPs were all significantly associated with precancerous cervical lesions.

In this study, the odds of developing PCL among women who did not have a formal education were four times higher than those who had formal education. This finding was consistent with other studies [153,154]. This could be explained as a lack of educational attainment might be linked with high-risk sexual behavior and a lack of information on STIs, including HPV, which is the leading cause of PCL and cervical cancer [155].

Moreover, the odds of developing precancerous cervical lesions among women living in rural areas were higher when compared to women living in urban areas. This finding was supported by other studies [156,157]. This is due to the fact that women had limited information and services access, poor health-seeking behavior, and lower coverage of HPV vaccination in rural as compared with urban areas [73,157]. The odds of developing precancerous cervical lesions were higher among women with multiple sexual partners compared to their counterparts [13,147,154]. Women with several sexual partners are clearly more vulnerable to STIs, particularly high-risk HPV strains, resulting in persistent HPV infection and precancerous cervical lesions [8,13].

In comparison to women without a history of STIs, those with a history of STIs were more likely to develop PCL. This finding is in agreement with other study findings [13,147,154]. This elevated risk is mainly associated with the exposure of high-risk HPV strains, primarily HPV 16 and 18, which account for nearly 70% of cervical cancer cases [158,159]. Furthermore, other STIs, such as chlamydia and herpes virus, could potentially contribute to cervical cancer progression by creating chronic inflammation, allowing HPV to persist and progress to PCL [160,161]. As a result, immunization against HPV at the optimal age of children and regular cervical cancer screening are crucial preventive measures to reduce this risk.

Moreover, initiation of early sexual intercourse before the age of 18 years was significantly associated with PCL. This conclusion is similar to other studies [162,163]. Early sexual activity is associated with an immature cervical epithelium

during adolescence, particularly the transformation zone (TZ), and prolonged cumulative exposure to HPV, the major cause of cervical cytological abnormalities [164,165]. It has also been linked to risky sexual behaviors, such as using condoms inconsistently or having several sexual partners [166].

The odds of developing precancerous cervical lesions were 3.33 times more likely among HIV-positive women when compared with those HIV-negative women. The finding of this study was supported by another systematic review and meta-analysis [144,149]. In general, HIV-infected women are much more likely to acquire precancerous cervical lesions due to the immunosuppression effect of HIV (low CD4 cells), higher risk of co-infections with other STIs, and persistent HPV infection [149,167]. This highlights the importance of primary HIV prevention, good ART adherence, integrating cervical cancer screening, and preventative strategies in HIV care programs in order to lower the risk of PCL and invasive cervical cancer.

In this study, women's CD4 counts were the other independent factor that was significantly associated with PCCL. Thus, HIV-positive women with CD4 counts below 200 cells/mm³ were five times more likely to develop PCL when compared to CD4 counts ≥200 cells/mm³. This conclusion is consistent with other study findings [13,15,149,168]. The possible explanation could be related to reactivation and persistent HPV infection due to immune deficiency or high viral load [15,161]. Therefore, prompt initiation and proper adherence to highly active antiretroviral therapy (HAART) is essential for boosting up the immune system and decreasing viral load, which ultimately reduce the occurrence of PCL and malignant cervical cancer.

Likewise, the odds of developing PCL were more likely among women who are not on ART than those on ART. This was in agreement with other studies, and this could be explained as a decrease in high-risk HPV persistence, and the histologic diagnosis of CIN2+ was linked to being on effective ART (i.e., individuals with longer duration, maintained HIV viral suppression, and steady higher CD4 cell count) [15,168,169]. This finding emphasizes the importance of ART in lowering the risk of PCL in HIV-positive women. It strengthens the immune system, accelerating the clearance of high-risk infections caused by HPV and decreasing the progression of cervical lesions, lowering the risk of advanced cervical cancer.

This study also concluded that women who smoked cigarettes were four times more likely to develop PCL than those who did not smoke cigarettes. This finding was congruent with prior meta-analysis studies [32,170,171]. A plausible explanation is that smoking exposes women to toxic chemicals that can directly damage cervical epithelium DNA. Additionally, cigarette smoking is strongly linked to aberrant methylation of tumor suppressor genes, such as p16 (CDNK2A), in women with high-grade CIN and squamous cell carcinoma [172,173]. Nicotine and other tobacco ingredients could harm the immune system's ability to get rid of HPV infections, which are highly linked to cervical neoplasia. This weakened defense allows for chronic HPV infections, raising the likelihood of CIN [174]. Therefore, health practitioners should always counsel women to undergo regular cervical cancer screening and abstain from smoking or quit smoking in order to prevent the long-term effects of tobacco.

The odds of developing precancerous cervical lesions were 4.39 times more likely among prolonged OCP users when compared with those who were non-users. This finding was congruent with prior meta-analysis studies [175–178]. The possible justification might be that prolonged use of oral contraceptives, particularly estrogen, may stimulate cervical epithelial proliferation, thereby making them more susceptible to HPV-induced DNA damage and neoplastic transformation [33,175,179]. So, regular cervical screening is recommended for women who use OCP for extended periods of time to ensure early detection and treatment of cervical abnormalities. Risk-benefit assessment on OCP and considering alternative contraception methods with healthcare professionals to reduce the risk of PCL.

## Implications of the study

This review highlights a high burden of precancerous cervical lesions among women undergoing cervical cancer screening across Africa. The pooled estimate indicates that a substantial proportion of screened women are at risk of progression to invasive cervical cancer if timely diagnosis and effective treatment interventions are not implemented. This finding

underscores the urgent need to strengthen national and regional efforts targeting both primary and secondary prevention of PCL and advanced cervical cancer. Thus, the findings of this review provide important baseline evidence for health professionals, policymakers, researchers, and global partners to design and implement evidence-based interventions to achieve the WHO 90-70-90 strategy targeted to eliminate CC. Key strategies should include expanding HPV vaccination coverage, scaling up accessible and cost-effective screening methods such as visual inspection with acetic acid and HPV DNA testing, and implementing population-based screening programs.

Furthermore, ensuring timely diagnosis and treatment of precancerous lesions is critical to interrupt disease progression. Integrating cervical cancer screening into existing reproductive healthcare services, such as HIV care and maternal health, can enhance coverage and efficiency. Strengthening health system readiness, including workforce training, provision of essential equipment, and infrastructure improvement alongside regular monitoring, evaluation, and follow-up mechanisms, is crucial to ensure continuity of care and long-term program effectiveness. These measures are vital for facilitating the early detection and management of precancerous cervical lesions, thereby assisting to lower the incidence of cervical cancer.

### Strengths and limitations of the study

This systematic review and meta-analysis synthesized evidence on the burden of precancerous cervical lesions and their associated factors in Africa, highlighting a critical public health issue. The findings serve as baseline evidence and offer valuable insights for clinicians, policymakers, researchers, and other stakeholders to prioritize the prevention and control of PCL and cervical cancer.

However, the following limitations should be considered when interpreting the findings of this review. Firstly, the findings of this review are based on previous primary studies that utilized calculated sample populations rather than nationally or regionally representative data. This may limit the generalizability of the finding to broader populations across African countries. Secondly, the presence of significant heterogeneity among the included studies might affect the inference to the continent. This might be due to variations in the study settings and methodologies, including variation in the sampling technique, sample size, analytical techniques, and difference in screening methods, which could affect the generalizability of the pooled estimate. This could be explained as the presence of substantial clinical and methodological heterogeneity across the studies resulting from the diverse in cervical cancer screening modalities utilized in the included studies specifically, VIA, cytology, and HPV DNA testing as well as the application of varying diagnostic algorithms (screen-and-treat, sequential, or combined approaches). These methods exhibit significant variations in sensitivity, specificity, and operator dependence result to diagnosis the presence of PCL.

Additionally, due to the concentration of studies in some countries, such as Ethiopia and Nigeria, it could restrict the conclusion of the finding to the continent. Another potential limitation of this review is the possibility of publication bias due to the use of PubMed as a major database for literature searches. Although PubMed is a well-known and extensive biomedical database, its sole or predominant use could have resulted in the exclusion of important studies indexed in other databases (for example, Embase, Scopus) or unpublished grey literature. To address this gap, we supplemented our search by systematically exploring Google Scholar to identify both published and unpublished studies, and we also used ScienceDirect to retrieve relevant articles from Elsevier-indexed journals.

Despite these limitations, the use of a random-effects model, along with subgroup and sensitivity analyses, strengthens the robustness of the findings and provides a valuable benchmark for future research and evidence-based policy development. Regarding the determinant factors, the confidence intervals for some variables, such as educational status, women's CD4 count, and smoking, were relatively wide, indicating potential uncertainty in the effect estimates and suggesting the need for cautious interpretation. This may be attributed to the small sample sizes in the primary studies and the limited number of studies included in the quantitative meta-analysis.

Finally, the estimated magnitude of PCL is primarily based on facility-based studies conducted across Africa. Consequently, the exact burden of PCL may be underestimated due to the limited inclusion of community-based data. To address this gap, we strongly recommend the expansion of both facility-based and community-based cervical cancer screening programs to enhance early detection and ensure timely treatment of PCL across African countries.

## Conclusions

In conclusion, the overall prevalence of precancerous cervical lesions in Africa is high (17%).

In this meta-analysis study, having no formal education, rural residency, a history of STIs, a history of multiple partners, early initiation of coitus, being an HIV-seropositive woman, a CD4 count of <200 cells/mm³, not being on ART, smoking, and prolonged use of oral contraceptive pills were significantly associated with precancerous cervical lesions. Therefore, health professionals, health administrators, and all other concerned bodies need to work in collaboration to expand comprehensive cervical cancer screening methods in healthcare facilities for early detection and treatment of cervical lesions. In addition, increasing community awareness and health education, expanding VIA in rural areas, expanding accessibility of screening methods, encouraging adherence to ART, overcoming risky sexual behaviors and practices, promoting safe sexual practices, providing special attention for high-risk groups (such as women living with HIV or having multiple sexual partners), and advocating early detection and treatment of precancerous cervical lesions.

## Supporting information

**S1 File. PRISMA 2020 Checklist.**
(DOCX)

**S2 File. Search strategies.**
(DOCX)

**S3 File. Excel spreadsheet for data extraction.**
(XLSX)

**S4 File. Sensitivity analysis.**
(DOCX)

## Author contributions

**Conceptualization:** Berihun Agegn Mengistie, Nuhamin Tesfa Tsega.

**Data curation:** Berihun Agegn Mengistie, Tazeb Alemu Anteneh, Kindu Yinges Wondie.

**Formal analysis:** Berihun Agegn Mengistie, Getie Mihret Aragaw, Alemneh Tadesse Kassie, Wondimnew Mersha Biset, Nuhamin Tesfa Tsega.

**Methodology:** Berihun Agegn Mengistie, Getie Mihret Aragaw, Alemken Eyayu Abuhay, Gebrye Gizaw Mulatu, Nuhamin Tesfa Tsega.

**Software:** Berihun Agegn Mengistie, Getie Mihret Aragaw, Alemneh Tadesse Kassie, Wondimnew Mersha Biset.

**Validation:** Berihun Agegn Mengistie, Getie Mihret Aragaw, Nuhamin Tesfa Tsega.

**Visualization:** Getie Mihret Aragaw, Alemken Eyayu Abuhay, Gebrye Gizaw Mulatu, Nuhamin Tesfa Tsega.

**Writing – original draft:** Getie Mihret Aragaw, Kindu Yinges Wondie, Alemken Eyayu Abuhay, Wondimnew Mersha Biset, Nuhamin Tesfa Tsega.

**Writing – review & editing:** Berihun Agegn Mengistie, Getie Mihret Aragaw, Tazeb Alemu Anteneh, Kindu Yinges Wondie, Alemneh Tadesse Kassie, Wondimnew Mersha Biset, Gebrye Gizaw Mulatu, Nuhamin Tesfa Tsega.

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
