## [Decision Letter · Decision Letter 0]

17 Jul 2025

Dear Dr. Mengistie,

Thank you for submitting your manuscript to PLOS ONE. After careful consideration, we feel that it has merit but does not fully meet PLOS ONE’s publication criteria as it currently stands. Therefore, we invite you to submit a revised version of the manuscript that addresses the points raised during the review process.

We look forward to receiving your revised manuscript.

Kind regards,

Kazunori Nagasaka

Academic Editor

PLOS ONE

Journal Requirements:

2. Please remove all personal information, ensure that the data shared are in accordance with participant consent, and re-upload a fully anonymized data set.

Additional guidance on preparing raw data for publication can be found in our Data Policy (https://journals.plos.org/plosone/s/data-availability#loc-human-research-participant-data-and-other-sensitive-data) and in the following article: http://www.bmj.com/content/340/bmj.c181.long .

Additional Editor Comments:

Dear Authors,

Thank you for submitting your manuscript to our journal.

After careful consideration, our decision is "Major Revision." Please thoroughly revise your manuscript in accordance with the reviewers' comments. When preparing your revised manuscript, ensure that you respond clearly and comprehensively to each comment, providing detailed explanations of the changes made or reasons for disagreement.

Submit your revision along with a point-by-point response letter clearly indicating the modifications and addressing all reviewer feedback.

We look forward to receiving your revised manuscript.

Sincerely,

Kazunori Nagasaka

Reviewers' comments:

Reviewer's Responses to Questions

**Comments to the Author**

1. Is the manuscript technically sound, and do the data support the conclusions?

Reviewer #1: Partly

Reviewer #2: Yes

2. Has the statistical analysis been performed appropriately and rigorously?

Reviewer #1: No

Reviewer #2: Yes

3. Have the authors made all data underlying the findings in their manuscript fully available?

Reviewer #1: Yes

Reviewer #2: Yes

4. Is the manuscript presented in an intelligible fashion and written in standard English?

Reviewer #1: Yes

Reviewer #2: Yes

Reviewer #1: This manuscript addresses an important public health concern in Africa. However, several aspects need improvement to meet the standard of a high-quality systematic review and meta-analysis, especially for publication in PLOS ONE. Below are my comments and suggestions:

Abstract:

1. Lines 31-34: The sentence is redundant and grammatically incorrect. Please revise it for clarity and conciseness.

2. Google Scholar and ScienceDirect are not considered formal scientific databases. They are scientific search engline. Please revise accordingly and clarify the actual databases used.

3. Please include the statistical methods used to estimate the pooled prevalence in the abstract.

4. The conclusion in the abstract is too lengthy. Kindly condense it to focus on the key findings and implications.

Introduction:

1. Since many countries already have cervical cancer care cascade data, please justify the rationale for conducting this study, especially with respect to screening data. What specific gap does this study aim to address?

2. The objectives section mentions identifying factors associated with precancerous cervical lesions. Please clarify why identifying these factors is important. Are there any previous systematic reviews on this topic? If so, how does this study build on or differ from them?

3. Please clearly justify the focus on African countries. What specific contextual or epidemiological reasons necessitate this geographical limitation?

4. The research question should be more clearly integrated with the stated objective(s) of the study.

Methods:

1. If the study used the CoCoPop framework (Condition, Context, Population) to define eligibility criteria, please state this explicitly. Each component should be clearly defined. For example, does the population include all women regardless of HIV status, age, or other criteria?

2. Please cite relevant tools/software appropriately. For instance, if EndNote 20 and STATA 17 was used, a proper citation or acknowledgment should be included.

3. Specify the statistical methods used to calculate the adjusted odds ratios (AORs) for associated factors.

Results:

1. Table 1 is not reader-friendly. The information is too dense. Please consider breaking it into clusters or themes for better clarity.

2. I believe the study design and mean age data are important to presented in Table 1.

3. Figure 1 (PRISMA flow diagram) needs improvement in terms of aesthetics and readability. Please revise using standard PRISMA templates.

4. Lines 223-224: Pelase cite included studies “In this 224 systematic review and meta-analysis, 112 eligible articles”

5. Table 4 would be more appropriate as a supplementary file to enhance the flow of the main manuscript.

6. Lines 266-267 and Table 5: One of the variables is reported with a wide confidence interval (AOR = 4.07, 95% CI: 1.74–9.53). This suggests high uncertainty or possible heterogeneity in the pooled effect estimate. In fact, many variables in the table show similarly wide intervals, suggesting that the pooled effect estimates may be unstable or based on small sample sizes or a limited number of studies. Please address this issue in the Discussion section by (1) Acknowledging the presence of wide confidence intervals across several variables. (2) Discussing potential reasons for this (e.g., variation in study design, measurement, sample size, or context). (3) Considering sensitivity or subgroup analyses to explore and explain the heterogeneity.

Discussion:

1. The discussion would benefit from further elaboration on the implications of the findings, especially regarding regional and national cervical cancer screening policies in Africa. How can policymakers or health systems use this evidence?

2. Given that PubMed appears to be the primary database searched, there is a risk of publication bias. This limitation should be explicitly discussed in the manuscript, including its potential influence on the findings.

Reference:

There appears to be a discrepancy in the reporting of included studies and referenced articles. You mentioned that 112 studies were included in the review, but only 72 studies are listed in the reference section.

Reviewer #2: Thank you for the opportunity to read this manuscript.

It is a very thorough and extensive approach to a very important topic, cervical cancer screening in LMI countries. As obvious from the included papers in the review and metaanalyses, the topic in complicated due to a number of different screening methods, varying study population size, HIV status over the huge continent.

Unfortunately, I have issues concerning the manuscript.

I think the introduction itself could be more condensed.

I do not see any clear definition of precancerous lesion, i.e. does it include only HSIL lesion in cytology, does it include other cytological diagnosis, does it mean the presence of hrHPV alone.

And foremost, which new knowledge compared to the state of the art does this manuscript ad?

Thus, in my opinion, the manuscript needs major revision before being considered for publication.

**Do you want your identity to be public for this peer review?** For information about this choice, including consent withdrawal, please see our Privacy Policy

Reviewer #1: **Yes: ** Sidik Maulana

Reviewer #2: No

---

## [Author Response · Author response to Decision Letter 1]

11 Aug 2025

Dear Academic editor and reviewers, Greetings!

I hope the authors' revision reaches you well. The authors are grateful for your constructive comments and suggestions to improve our manuscript. As a result, we diligently address all the raised comments in the manuscript with tracked changes and in the updated version of the manuscript. In addition, we upload the point-by-point response to reviewers in a separate file. We have uploaded all the relevant files. We are waiting to see your feedback.

Respectfully,

Berihun Agegn, corresponding author

---

## [Decision Letter · Decision Letter 1]

8 Sep 2025

Dear Dr. Mengistie,

Thank you for submitting your manuscript to PLOS ONE. After careful consideration, we feel that it has merit but does not fully meet PLOS ONE’s publication criteria as it currently stands. Therefore, we invite you to submit a revised version of the manuscript that addresses the points raised during the review process.

We look forward to receiving your revised manuscript.

Kind regards,

Kazunori Nagasaka

Academic Editor

PLOS ONE

Journal Requirements:

**Additional Editor Comments:**

Dear Authors,

Thank you for re-submitting your revised manuscript to PLOS ONE.

After review, our reviewers have suggested that the manuscript requires further revision.

We kindly ask you to revise the manuscript with a more in-depth discussion to fully address their comments and suggestions.

We look forward to receiving your revised manuscript.

Sincerely,

Kazunori Nagasaka

Reviewers' comments:

Reviewer's Responses to Questions

**Comments to the Author**

Reviewer #1: All comments have been addressed

Reviewer #2: (No Response)

2. Is the manuscript technically sound, and do the data support the conclusions?

Reviewer #1: Yes

Reviewer #2: Partly

3. Has the statistical analysis been performed appropriately and rigorously?

Reviewer #1: Yes

Reviewer #2: Yes

4. Have the authors made all data underlying the findings in their manuscript fully available?

Reviewer #1: Yes

Reviewer #2: Yes

5. Is the manuscript presented in an intelligible fashion and written in standard English?

Reviewer #1: Yes

Reviewer #2: Yes

Reviewer #1: Thank you for your thoughtful revisions. We truly appreciate the improvements you made and are pleased with the updated version of the manuscript. As this meta-analysis involves many included studies, we kindly suggest carefully checking for any errors during the next stage, especially in citations and references, since some issues may arise during the galley proof stage. We belive that ensuring accuracy is the authors’ responsibility.

Reviewer #2: I have with interest taken part of the revised manuscript.

The area of the paper is still, or even now more, important in the light of recent developments concerning the availability concerning antiviral therapy in SSA and its potential of an even more critical situation in cervical cancer morbidity and mortality.

However, some of my concerns still prevail.

Concerning the introduction, it would benefit from a more comprehensive and structured approach.

As a minor comment PCCL is in a few places used instead of PCL in the manuscript.

Concerning especially the meta analyses, I have concerns concerning the lack of diagnostic stringency (ref 49) in at least one of the rather few original articles I have read myself, and as far as I understand LSIL is included in the PCL outcome of other studies (ref 51) whilst only histological CIN2+ after VIA and/or HPV screening seems to be the included outcome concerning ref 53, however the figures might need to be checked. My point is not some minor possible error in the statistical material, but the problem doing meta-analyses on such a complex material with different screening methods (VIA, HPV, cytology) and different approaches using these alone or in combination, is it even valid in the medical perspective.

Thus, at least this must be further and in depth discussed in the limitations of the study.

**Do you want your identity to be public for this peer review?** For information about this choice, including consent withdrawal, please see our Privacy Policy

Reviewer #1: **Yes: ** Sidik Maulana

Reviewer #2: No

---

## [Author Response · Author response to Decision Letter 2]

20 Oct 2025

Dear Academic Editor and Reviewers, we really appreciate you all for your timely feedback and constructive comments, which are vital for improving our manuscript. Thus, we carefully addressed the comments as much as possible. We kindly and respectfully invite to review the the separate file "Response to Reviewers." We look forward to your kind response.

Respectfully,

---

## [Decision Letter · Decision Letter 2]

24 Nov 2025

Prevalence and determinants of precancerous cervical lesions among women screened for cervical cancer in Africa: a systematic review and meta-analysis

PONE-D-25-16143R2

Dear Dr. Mengistie,

We’re pleased to inform you that your manuscript has been judged scientifically suitable for publication and will be formally accepted for publication once it meets all outstanding technical requirements.

Kind regards,

Kazunori Nagasaka

Academic Editor

PLOS ONE

Additional Editor Comments (optional):

Dear Authors,

Thank you very much for submitting the revised version of your manuscript.

I have carefully reviewed your detailed responses, the updated manuscript, and the incorporated revisions.

I am pleased to confirm that both reviewers have recommended acceptance, and all previous concerns have been fully addressed.

The manuscript shows substantial improvement in clarity, structure, and methodological transparency.

Your thorough responses, particularly regarding diagnostic variability, study heterogeneity, and the rationale for including different screening methods, are appreciated and have strengthened the overall quality of the paper.

Reviewer #1 expressed satisfaction with the revisions and emphasized the importance of double-checking citations and references at the proof stage.

Reviewer #2 acknowledged the improvements and highlighted the value of this work given the evolving landscape of cervical cancer prevention in sub-Saharan Africa.

Although the reviewer reiterated concerns about methodological heterogeneity, your expanded discussion adequately addresses these limitations.

After this long review process, I am pleased to inform you that your manuscript is accepted for publication in PLOS ONE.

Thank you for your thoughtful revisions and your contribution to the field.

We look forward to seeing your work published.

Sincerely,

Kazunori Nagasaka

Reviewers' comments:

Reviewer's Responses to Questions

**Comments to the Author**

Reviewer #2: (No Response)

2. Is the manuscript technically sound, and do the data support the conclusions?

Reviewer #2: (No Response)

3. Has the statistical analysis been performed appropriately and rigorously?

Reviewer #2: (No Response)

4. Have the authors made all data underlying the findings in their manuscript fully available?

Reviewer #2: (No Response)

5. Is the manuscript presented in an intelligible fashion and written in standard English?

Reviewer #2: (No Response)

Reviewer #2: (No Response)

**Do you want your identity to be public for this peer review?** For information about this choice, including consent withdrawal, please see our Privacy Policy

Reviewer #2: No

---

## [Editor Report · Acceptance letter]

1 Dec 2025

PONE-D-25-16143R2

PLOS ONE

Dear Dr. Mengistie,

I'm pleased to inform you that your manuscript has been deemed suitable for publication in PLOS ONE. Congratulations! Your manuscript is now being handed over to our production team.

Kind regards,

on behalf of

Professor Kazunori Nagasaka

Academic Editor

PLOS ONE